

# Comparative gene expression analysis of *Beauveria bassiana* against *Spodoptera frugiperda*

Hamdy H. Aly[1,2], Yun Meng[1] and Dun Wang[1]

[1] Institute of Entomology, Northwest A&F University, Yangling, Shaanxi, China
[2] Desert Research Center, Cairo, Egypt

## ABSTRACT

**Background.** This study investigates the gene expression dynamics and biocontrol effectiveness of *Beauveria bassiana* against *Spodoptera frugiperda*, the fall armyworm, a notable agricultural pest. Our objectives were to analyze the *B. bassiana* gene expression variation during insect infection compared to grow on artificial media and to evaluate the effects of different spore concentrations on larval mortality, development, and reproduction.

**Methods.** A combination of bioassays and transcriptome analysis was employed. *S. frugiperda* larvae were exposed to different spore concentrations, and mortality rates were recorded at various developmental stages. RNA sequencing was performed on fungal samples from infected larvae and those grown on 1/4-strength Sabouraud Dextrose Agar with Yeast Extract (SDAY) media. Differential gene expression libraries were constructed at 48, 96, and 144 hours' post-infection. Gene Ontology (GO) and Kyoto Encyclopedia of Genes and Genomes (KEGG) pathway analyses were used to identification of biological processes and pathways that differentiate infection from growth on artificial media.

**Results.** The highest spore concentration ($1 \times 10^7$ spores/mL) significantly increased larval mortality, prolonged developmental stages, and reduced reproductive success, particularly in pupation, adult emergence, and female fecundity. Transcriptomic analysis revealed substantial differences in gene expression between *B. bassiana* grown on artificial media and during host infection at three-time points. At 48 hours' post-infection, genes involved in adhesion and cuticle penetration, such as serine/threonine-protein kinases (STPKs) and lipases, were upregulated, indicating adaptation to host invasion. GO analysis revealed enrichment in cellular and catalytic activities, while KEGG pathways highlighted early-stage metabolic adaptations related to nutrient acquisition and energy metabolism. In contrast, fungal growth on artificial media showed minimal expression of infection-related genes. At 96 hours, genes associated with ABC transporters and detoxification were significantly upregulated, supporting fungal survival and immune evasion. GO terms were enriched in membrane components, and KEGG pathways focused on energy metabolism and stress responses. At 144 hours, genes related to secondary metabolism were upregulated, indicating the production of compounds vital for continued invasion and immune suppression. The activation of these pathways were minimal or absent during growth on artificial media.

**Conclusions.** This study provides new insights into the molecular adaptations of *B. bassiana* during host infection, revealing key virulence factors and infection dynamics. The identified gene expression signatures enhance our understanding of

Corresponding author
Dun Wang, wang-hande@nwsuaf.edu.cn

fungal infection mechanisms and could inform more effective biocontrol strategies for managing agricultural pests.

## INTRODUCTION

The fall armyworm (FAW), *Spodoptera frugiperda* (J.E. Smith) (Lepidoptera: Noctuidae), is a highly destructive agricultural pest with a broad host range, causing severe economic losses in staple crops such as maize, rice, sorghum, and wheat (*Diagne et al., 2021*; *Padhee & Prasanna, 2019*; *Yang et al., 2020*). Native to the Americas, it has rapidly spread to Africa, Asia, and other regions, posing a significant threat to global food security due to its strong migratory ability, rapid reproduction, and resistance to conventional chemical pesticides (*Goergen et al., 2016*; *Kalleshwaraswamy et al., 2018*). A limitations of chemical control strategies, biological control agents such as entomopathogenic fungi, including *Beauveria bassiana* Bals. (Vuil.), have gained considerable attention as sustainable alternatives for managing *S. frugiperda* populations (*Abdel Galil et al. 2019*; *Montezano et al., 2018*).

*B. bassiana* is a versatile entomopathogenic fungus capable of infecting a broad range of insects, arachnids, and nematodes while also living asymptomatically within plants as an endophyte (*Jensen et al., 2020*). Its virulence naturally varies across a host range of over 700 insect species, generating significant interest in studying *B. bassiana*-host interactions (*Xiang et al., 2024*). The infection process involves critical stages, including initial adhesion, conidia germination, appressorium formation, and cuticle degradation facilitated by enzymes such as proteases and chitinases (*Ortiz-Urquiza, 2021*; *Silva et al., 2020*). Once fungus proliferates inside the host as blastospores, evades immune detection, releases toxins, and spreads through the hemolymph, ultimately leading to host death (*Wang et al., 2017*). The precise temporal and spatial expression of numerous genes is crucial at various infection stages (*Qiu et al., 2015*; *Valero-Jiménez et al., 2016*). Fungal pathogens must adapt to distinct environments to successfully infect their hosts. *B. bassiana* undergoes significant physiological and molecular changes when transitioning from saprophytic growth on artificial media to a parasitic phase within a host insect. This transition involves alterations in gene expression that are essential for infection, survival, and reproduction. High-throughput methods are frequently employed to identify genes involved in host-pathogen interactions. While previous research identified over 2,000 genes under different growth conditions *via* expressed sequence tag (EST) analysis (*Mantilla et al., 2012*), a comprehensive understanding of global gene expression in response to host environments remains limited because of earlier methodological constraints. RNA-seq technology offers a revolutionary approach for analyzing gene expression with high resolution and detail, addressing previous limitations (*Guo et al., 2024*). The published genome of *B. bassiana* serves as a valuable reference for transcriptome analyses to uncover genetic mechanisms (*Wang et al., 2017*). Transcript profiling through RNA-seq has proven

effective in identifying gene sets that are differentially expressed under specific conditions (*Stupnikov et al., 2021*).

Recent transcriptomic studies of entomopathogenic fungi, including *B. bassiana,* have revealed critical insights into gene regulation during host infection. For instance, transcriptome analyses of *Metarhizium rileyi* have identified genes involved in cuticle degradation (*Fan et al., 2023*). Similarly, comparative transcriptomic studies of *B. bassiana* have identified differentially expressed genes during the infection of various insect hosts, shedding light on pathogenicity-related pathways, such as those regulating cuticle penetration, stress response, and host adaptation (*Chen et al., 2018a*; *Mantilla et al., 2012*). While these studies have provided valuable insights into the gene expression of *B. bassiana*, they have predominantly examined individual host species or specific growth conditions. This narrow focus has resulted in a limited comparative understanding of how *B. bassiana's* gene expression differs between infection of hosts and saprophytic growth.

This study aims to bridge this gap by employing RNA-seq transcriptomic analysis to compare gene expression profiles at various time-interval of *B. bassiana* infection of *S. frugiperda versus* growth on artificial media. By identifying key genes and pathways involved in these processes, we aim to enhance our understanding of how *B. bassiana* adapts its biological functions to different environments. Additionally, this study investigated the impact of various spore concentrations on *S. frugiperda* larvae, assessing mortality, developmental delays, and reproductive outcomes. This multifaceted approach, which combines transcriptomic analysis and biological assays, aims to uncover the molecular adaptations of *B. bassiana* during host infection, contributing to our understanding of fungal pathogenicity and laying a foundation for future studies on fungal biology and host adaptation.

## MATERIALS & METHODS

### Insect rearing
The eggs of *S. frugiperda* used in this study were sourced from larvae collected from maize fields in Guangdong Province, China, during the maize growing season. In accordance with *Ge et al. (2022)*, the collected larvae were cultured on a specially formulated artificial diet. The diet consisted of the following components: wheat bran (50 g), soybean powder (40 g), maize powder (100 g), yeast powder (30 g), agar (24 g), casein (40 g), sorbic acid (three g), ascorbic acid (3.5 g), vitamins (0.15 g), formaldehyde (four mL), glacial acetic acid (four mL), and distilled water (1,200 mL). This mixture provided the necessary nutrients for larval development and ensured consistent growth conditions. The resulting adult *S. frugiperda* were provided with artificial diets containing a 10% honey solution soaked in sterile cotton balls. In the laboratory, the larvae were maintained under controlled conditions at a temperature of $25 \pm 2$ °C, a photoperiod of 12:12 (dark: light), and a relative humidity of $65 \pm 5$%. After laying their eggs, the batches were carefully transferred and placed in an environmental growth chamber that maintained the laboratory conditions. When they hatched, the newborn larvae were moved to transparent rectangular plastic boxes measuring $28 \times 17 \times 18$ cm.

### Investigating the insecticidal potential of *B. bassiana* CDL1 against third-instar *S. frugiperda* larvae

The entomopathogenic fungus *B. bassiana* CDL1 was cultivated on 1/4-strength Sabouraud dextrose agar with yeast extract (SDAY) medium at a temperature of 25 °C for 14 days. Afterward, conidia were collected from the fungus and suspended in a solution containing 0.01% Tween-20; Tween-20 was added as a surfactant to reduce surface tension, ensure even dispersion of the fungal spores, and prevent clumping, thereby improving the uniformity of spore coverage during application and ensuring accurate concentrations for reliable experimental results to achieve a concentration of $3 \times 10^8$ conidia/mL. This initial conidial suspension served as a stock mixture, and subsequent dilutions were made to obtain concentrations of $1 \times 10^4$, $1 \times 10^5$, $1 \times 10^6$, and $1 \times 10^7$ spores/mL. For insecticidal activity, the third-instar larvae of *S. frugiperda* were exposed to different concentrations of fungal spores *via* a spray method. Each larva was treated with 0.5 mL of the spore suspension applied uniformly using a handheld sprayer calibrated to deliver a fine mist. The sprayer was held at a consistent distance of 15 cm from the larvae to ensure even coverage, and the nozzle was adjusted to produce droplets of approximately 50–100 µm in diameter. This method ensured uniform spore distribution over the larval surface and minimized variability in spore deposition. Control groups were sprayed with sterile water containing 0.5% glycerin and 0.01% Tween-20. Larval mortality was recorded daily, and dead larvae were surface sterilized before being examined for the presence of fungal hyphae and conidia to confirm their death (*Domingues et al., 2022*). Parameters such as larval duration, mortality rate, pupation rate, pupal duration, pupal mortality, and adult deformities were recorded. Fecundity, deficient fecundity percentage, and the oviposition deterrent index were calculated for the emerging adults according to the following equations:

**A.** Fecundity = Number of eggs laid per female
**B.** Deficient fecundity $= \frac{\text{Control} - \text{Treated}}{\text{Control}} \times 100$
**C.** Oviposition deterrent index $= \frac{\text{C} - \text{T}}{\text{C} + \text{T}} \times 100$

where C and T are the mean numbers of eggs laid in the control and *B. bassiana* CDL1-treated larvae, respectively (*Huang, Renwick & Chew, 1994*). To analyze the mortality data, Abbott corrections were applied, and probit analysis was conducted to determine the mean lethal concentration ($LC_{50}$) following established protocols (*Abbott, 1925*; *Finney, 1971*). This study included observations up to the adult emergence stage, which offered a holistic perspective on the insecticidal efficacy of *B. bassiana* CDL1.

### Transcriptomic profiling of the entomopathogenic fungus *B. bassiana* strain CDL1: investigation of pathogenicity and gene expression dynamics during *S. frugiperda* infection

This study aimed to analyze the transcriptome of entomopathogenic fungus *B. bassiana* strain CDL1, which was selected for its high efficacy in causing larval mortality. The fungus was cultivated in 50 ml of 1/4-strength SDAY broth medium under controlled conditions: shaking at 145 rpm/min, shaking at 24 °C, and a pH of 6.6. For inoculation, a 0.5-mL culture containing approximately $1 \times 10^8$ conidia/mL was used. The choice of 1/4-strength

SDAY broth media allowed for the simulation of nutrient-limited conditions, which more closely resemble the environment of fungus encounters during infection, thus enhancing the expression of genes related to pathogenicity and stress responses. Additionally, this nutrient limitation reduces the overexpression of growth-related genes, ensuring a clearer focus on infection-relevant processes (*Schumann, Smith & Wang, 2013*; *Zhang et al., 2021*). To assess fungal pathogenicity, third-instar *S. frugiperda* larvae were infected with the same spore suspension until they reached the pupal stage. All samples were collected at 48, 96, and 144 hours' post-infection to capture distinct transcriptomic profiles across infection stages, with three biological replicates taken for each time point, both from fungal culture in broth media and infected larvae, individually. Immediately after collection, all samples were snap-frozen in liquid nitrogen for five minutes to preserve RNA integrity and stored at −80 °C until RNA extraction.

## Total RNA extraction, library construction, and sequencing

Total RNA was extracted using the TRIzol reagent (Invitrogen, Carlsbad, CA, USA), following the manufacturer's instructions, with adaptations for fungal growth in 1/4-strength SDAY broth medium and fungus-infected larvae. For each condition and time point (48 h, 96 h, and 144 h), three biological replicates were prepared to ensure statistical reliability and capture biological variability. The quality of the RNA was assessed using an Agilent 2100 Bioanalyzer (Agilent Technologies, Palo Alto, CA, USA) and RNase-free agarose gel electrophoresis. Eukaryotic mRNA was enriched using oligo(dT) beads, while prokaryotic mRNA was enriched by removing rRNA with the Ribo-Zero™ Magnetic Kit (Epicenter, Madison, WI, USA). The enriched mRNA was fragmented into short pieces using fragmentation buffer and reverse transcribed into cDNA using the NEBNext Ultra RNA Library Prep Kit for Illumina (NEB #7530; New England Biolabs, Ipswich, MA, USA). The resulting double-stranded cDNA fragments underwent end repair, A-base addition, and ligation to Illumina sequencing adapters. The ligation reaction mixture was purified using AMPure XP beads (1.0X) and subjected to polymerase chain reaction (PCR) amplification. The final cDNA libraries were sequenced on the Illumina NovaSeq 6000 platform. Each library was sequenced to a depth ranging from 36.1 million to 49.5 million reads per sample, ensuring sufficient coverage for accurate gene expression and differential expression analysis. Sequencing was conducted by Gene Denovo Biotechnology Co. (Guangzhou, China) (https://www.genedenovo.com). Raw sequencing data underwent quality control to remove low-quality reads, and the resulting clean data were used for downstream analyses.

## Mapping of RNA-seq reads and quantitative analysis of gene expression

The raw reads in FASTQ format were filtered *via* FASTP (*Chen et al., 2018b*) (version 0.18.0) to obtain clean data. Quality statistics, including the Q20, Q30, and GC content, were calculated. The resulting clean reads were then mapped to the reference genome of *B. bassiana* (*Xiao et al., 2012*) *via* HISAT2 (v2.1.0) (*Kim et al., 2019*). To count the number of reads mapped to each gene, Gffcompare (version 0.12.6) (*Pertea & Pertea,*
*2020*) was used, and the FPKM value was calculated. The power analysis calculation (alpha = 0.05, effect size = 2) was carried out on all the genes of the triplicate cultures of *B. bassiana* grown in 1/4-strength SDAY broth medium and fungus-infected larvae, using the transcript per million (TPM) values as sequencing depth. The analysis was conducted (online at https://rodrigo-arcoverde.shinyapps.io/rnaseq_power_calc/), which confirmed sufficient statistical power for detecting differentially expressed genes (DEGs).

The software DESeq2 (*Love, Huber & Anders, 2014*) was used to conduct differential expression analysis of RNAs between six different groups. Genes with a false discovery rate (FDR) <0.05 and an absolute fold change ≥ 2 were considered differentially expressed genes (DEGs).

## GO functional classification and KEGG pathway enrichment analysis of DEGs

To further identify the functions of the differentially expressed genes, the GO and KEGG enrichment analyses were done. For GO enrichment analysis, the Gene Ontology database (https://geneontology.org/) was used. To ensure accuracy, all *p*-values were subjected to Bonferroni correction (*Abdi, 2007*) and a corrected *p*-value of <0.05 as the threshold for significant enrichment of the gene sets were considered. Additionally, the Kyoto Encyclopedia of Genes and Genomes (KEGG) pathway enrichment analysis for DEGs *via* the clusterProfiler package (*Yu et al., 2012*) was performed.

## RT-qPCR validation of differentially expressed genes identified *via* RNA-seq

To validate the differentially expressed genes (DEGs) identified through RNA-seq analysis, we selected a total of 18 DEGs for validation *via* real-time quantitative PCR (RT-qPCR). Initially, 10 DEGs were randomly selected from the pool of statistically significant DEGs (adjusted *p*-value < 0.05 and |log2FC| > 1) to provide a broad validation of the RNA-seq results. Additionally, we included eight more DEGs that were specifically chosen for their biological relevance and involvement in key pathways or processes related to the study's objectives. This combined approach ensures both a representative validation of the RNA-seq findings and a targeted analysis of functionally important genes. RT-qPCR was performed using an iCycler iQ Real-time PCR System (Bio-Rad, Hercules, CA, USA) with the QuantiNove SYBR Green PCR Kit (Vazyme Biotech Co., Ltd., China), following the manufacturer's instructions. The cycling parameters were as follows: initial denaturation at 95 °C for 10 s, followed by 40 cycles of 95 °C for 10 s, 56.5 °C for 20 s, and 72 °C for 20 s. A melting curve analysis was performed at the end of each run to confirm the specificity of the amplified products. To normalize the expression levels of the target DEGs, we used the expression of 18S rRNA as an internal reference gene. Relative gene expression was calculated using the $2^{-\Delta\Delta Ct}$ method. The primers used for RT-qPCR were designed using Primer-BLAST (NCBI) and are listed in Table S1. All reactions were performed in triplicate to ensure technical reproducibility.
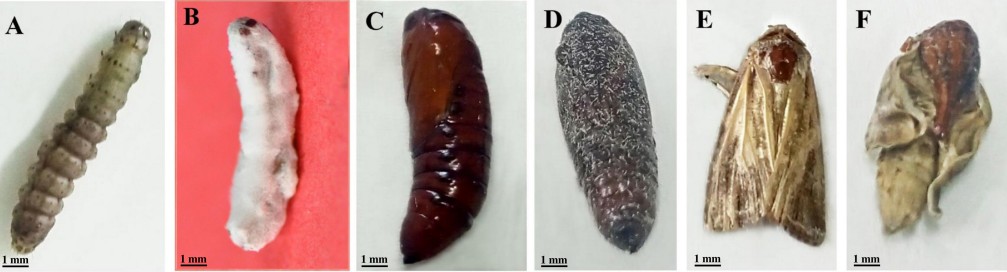

**Figure 1 Developmental stages of *S. frugiperda* infected by *B. bassiana* strain CDL1: a morphological perspective.** (A) Normal larvae; (B) cadaver formation of larvae due to *B. bassiana* (CDL1) infection; (C) normal pupae; (D) abnormal pupae with distinct dark coloration infection; (E) normal adult; (F) deformed adult.

## Statistical analysis

The bioassay experiments were conducted in triplicate to improve reliability. The analyses included an independent $t$-test and one-way ANOVA, followed by Duncan's multiple range test. The statistical program SPSS was used for these analyses. The results are presented as the means $\pm$ SE, and a significance level of $p < 0.05$ was considered statistically significant. Additionally, a heatmap illustrating the concentration-dependent effects of *B. bassiana* CDL1 on various parameters of *S. frugiperda* was generated using OriginLab version 2024.

## RESULTS

### The impact of *B. bassiana* CDL1 spores on mortality rates and developmental progression at the third larval instar of *S. frugiperda*

This study investigated the impact of different concentrations of *B. bassiana* CDL1 spores on the mortality rates of *S. frugiperda*. The results revealed that the mortality of both larvae and pupae was positively correlated with increasing concentrations of *B. bassiana* CDL1 spores. In particular, larval mortality increased from $19.99 \pm 1.72\%$ at the lowest concentration of $1 \times 10^4$ spores/mL to $66.66 \pm 1.72\%$ at the highest concentration of $1 \times 10^7$ spores/mL, demonstrating the efficacy of the treatment. Pupal mortality also increased, reaching $32.77 \pm 1.94\%$ at the highest concentration. The transition from larvae to adults was significantly impeded, with a failure rate of $99.44 \pm 0.24\%$ at the highest concentration. The corrected percentage of individuals who failed to develop into adults reached $95.05 \pm 1.10\%$ at the highest concentration when accounting for natural mortality (Fig. 1 and Table 1). The results of the probit analysis for the estimation of the $LC_{50}$ value for mortality in *S. frugiperda* were $3.83 \times 10^4$ spores/mL (Fig. S1). The $LC_{50}$ values clearly revealed that *B. bassiana* had considerable toxic effects on *S. frugiperda*.

### Effects of different concentrations of *B. bassiana* CDL1 spores on the developmental parameters of *S. frugiperda* during the third larval instar

The findings presented in Table 2 shed light on the influence of spore concentration on crucial developmental stages. Notably, the duration of larval development, measured in

**Table 1  Mortality rates of fall armyworm (*S. frugiperda*) at different developmental stages after treatment with various concentrations of *B. bassiana* CDL1 spores.**

| Conc. (spore/mL) | Larval mortality (%) | Pupal mortality (%) | Individuals failed to develop to adults (observed) (%) | Individuals failed to develop to adults (corrected) (%) |
|---|---|---|---|---|
| Control | $8.88 \pm 0.99^a$ | $0.00 \pm 0.00^a$ | $8.88 \pm 0.99^a$ | $0.00 \pm 0.00^a$ |
| $1 \times 10^4$ | $19.99 \pm 1.72^{ab}$ | $24.86 \pm 1.99^b$ | $44.84 \pm 2.18^b$ | $41.57 \pm 1.37^b$ |
| $1 \times 10^5$ | $31.11 \pm 2.62^b$ | $24.57 \pm 2.02^b$ | $55.68 \pm 3.95^b$ | $52.50 \pm 3.87^b$ |
| $1 \times 10^6$ | $62.22 \pm 0.99^c$ | $26.85 \pm 1.49^b$ | $89.07 \pm 1.07^c$ | $87.72 \pm 1.15^c$ |
| $1 \times 10^7$ | $66.66 \pm 1.72^c$ | $32.77 \pm 1.94^b$ | $99.44 \pm 0.24^c$ | $95.05 \pm 1.10^c$ |

Notes.
Data are presented as the mean value ± SE. Means in the same columns followed by the same letters are not significantly different.

**Table 2  Shows the effects of varying concentrations of *B. bassiana* CDL1 spores on different parameters of the fall armyworm, which was treated as the third larval instar.**

| Conc. (spore/mL) | Larval duration (Mean days ±) | Pupation (%) | Pupal duration (Mean days ±) | Adult emergence (%) | Adult longevity (Mean days ±) |
|---|---|---|---|---|---|
| Control | $10.03 \pm 0.09^a$ | $91.11 \pm 0.94^a$ | $9.33 \pm 0.14^a$ | $91.13 \pm 0.99^a$ | $10.26 \pm 0.24^a$ |
| $1 \times 10^4$ | $12.99 \pm 0.08^a$ | $80.00 \pm 1.72^{ab}$ | $9.66 \pm 0.39^{ab}$ | $55.15 \pm 2.81^b$ | $2.00 \pm 0.25^b$ |
| $1 \times 10^5$ | $13.86 \pm 0.31^b$ | $68.88 \pm 2.62^b$ | $11.00 \pm 0.25^{ab}$ | $44.31 \pm 3.95^b$ | $1.66 \pm 0.14^b$ |
| $1 \times 10^6$ | $14.39 \pm 0.59^b$ | $37.78 \pm 0.99^c$ | $11.66 \pm 0.39^b$ | $10.93 \pm 1.07^c$ | $1.33 \pm 0.14^b$ |
| $1 \times 10^7$ | $14.86 \pm 0.26^b$ | $33.33 \pm 1.72^c$ | $14.33 \pm 0.14^c$ | $0.56 \pm 0.14^c$ | $1.33 \pm 0.14^b$ |

Notes.
Data are presented as the mean value ± SE. Means in the same columns followed by the same letters are not significantly different.

mean days, consistently increased with corresponding concentrations of *B. bassiana* CDL1 spores. The control group exhibited a development period of $10.03 \pm 0.09$ days, which extended significantly to $14.86 \pm 0.26$ days at the highest spore concentration. Moreover, the pupation rate decreased to $33.33 \pm 1.72\%$ at the highest spore concentration, compared to the control group, which had a pupation rate of $91.11 \pm 0.94\%$. The pupal duration followed a comparable trend, extending from $9.33 \pm 0.14$ days in the control group to $14.33 \pm 0.14$ days at the highest spore concentration. In parallel, adult emergence was severely affected, decreasing dramatically from $91.11 \pm 0.94\%$ in the control group to just $0.56 \pm 0.14\%$ at the highest concentration. Finally, adult longevity was significantly reduced, with lifespan declining from $10.26 \pm 0.24$ days in untreated individuals to only $1.33 \pm 0.14$ days in those exposed to the highest concentration.

## The impact of *B. bassiana* CDL1 spores on the reproductive parameters of *S. frugiperda* in the third larval instar

Different concentrations of *B. bassiana* CDL1 affect the reproductive parameters of *S. frugiperda* in their third larval stage. The concentration range varied from the control group to $1 \times 10^7$ spores/mL, revealing clear trends in the sex ratio, fecundity, deficient fecundity, and oviposition deterrent index (Table 3). The sex ratio, representing the percentage of females in the population, showed a steady increase with rising spore concentrations, starting at $0.47 \pm 0.12$ in the control group and reaching $1.33 \pm 0.14$ at

**Table 3** Illustrates the impact of varying concentrations of *B. bassiana* CDL1 spores on the fecundity and sex ratio of the fall armyworm, which was treated as the third larval instar.

| Conc. (spore/mL) | Sex ratio (Female/total) | No. of egg/female (Fecundity) $\pm$ SE | Deficient fecundity (%) | Oviposition deterrent index (%) |
|---|---|---|---|---|
| **Control** | $0.47 \pm 0.12^a$ | $376.66 \pm 6.49^a$ | $0.00 \pm 0.00^a$ | $0.00 \pm 0.00^a$ |
| $1 \times 10^4$ | $0.44 \pm 0.03^a$ | $69.33 \pm 2.84^b$ | $81.62 \pm 0.61^b$ | $68.99 \pm 0.86^b$ |
| $1 \times 10^5$ | $0.47 \pm 0.03^a$ | $32.00 \pm 1.86^c$ | $88.07 \pm 1.37^c$ | $83.89 \pm 0.98^c$ |
| $1 \times 10^6$ | $0.54 \pm 0.04^a$ | $0.00 \pm 0.00^d$ | $100.00 \pm 0.00^d$ | $100.00 \pm 0.00^d$ |
| $1 \times 10^7$ | $1.33 \pm 0.14^b$ | $0.00 \pm 0.00^d$ | $100.00 \pm 0.00^d$ | $100.00 \pm 0.00^d$ |

**Notes.**
Data are presented as the mean value $\pm$ SE. Means in the same columns followed by the same letters are not significantly different.

the highest concentration. Fecundity, measured as the number of eggs laid per female, declined significantly as spore concentration increased. The control group exhibited a high fecundity rate of $376.66 \pm 6.49$ eggs per female. However, at a concentration of $1 \times 10^6$ spores/mL, egg production ceased entirely. This was reflected by a deficient fecundity rate of $100.00 \pm 0.00\%$, indicating a complete inhibition of reproduction. The oviposition deterrent index, indicating the effectiveness of *B. bassiana* CDL1 in deterring oviposition, tends to increase with increasing spore concentration. The index reached $100.00 \pm 0.00\%$ at $1 \times 10^6$ spores/mL, indicating a complete deterrent effect on oviposition at the highest concentrations tested.

## Impact of concentration-dependent *B. bassiana* on life stages of *S. frugiperda*: a heatmap analysis

This study provides a detailed examination of the effects of *B. bassiana* CDL1 on various life stages of *S. frugiperda* through a concentration-dependent heatmap analysis. The heatmap in Fig. 2 visually represents the impacts of different spore concentrations of *B. bassiana* CDL1 on key biological parameters, including larval and pupal mortality, developmental duration, fecundity, and sex ratio. The numbers within the heatmap circles indicate the percentage of larvae and pupae that died at each concentration, showing that higher concentrations of *B. bassiana* resulted in increased mortality, emphasizing its potential efficacy as a biocontrol agent. Additionally, the heatmap reveals that elevated spores concentrations extended the duration of the larval and pupal stages, indicating a disruption of normal development and may inducing physiological stress caused by infection. Reductions in fecundity were observed as spore concentrations increased, with fewer eggs laid by adult females, while the sex ratio data show a shift in the proportion of males to females, implying potential alterations in population structure. The color gradients of the heatmap highlight the intensity of these effects, with darker shades corresponding to greater impacts, providing a comprehensive visual overview of the multifaceted interactions between *B. bassiana* CDL1 and various life cycle parameters of *S. frugiperda*. This analysis demonstrates both lethal and sublethal effects of fungal infection on developmental and reproductive outcomes.

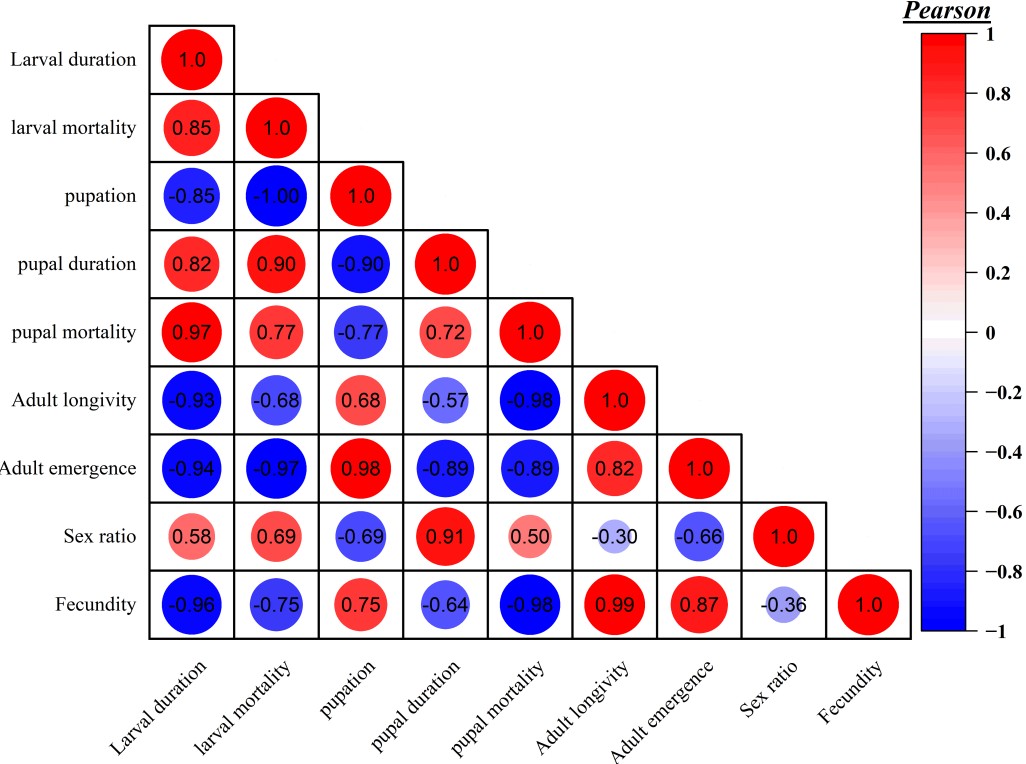

**Figure 2** Heatmap analysis reveals concentration-dependent effects of *B. bassiana* CDL1 on various parameters of the fall armyworm (*S. frugiperda*).

## Effects of *B. bassiana* CDL1 on *S. frugiperda* at different time points

The virulence of the *B. bassiana* CDL1 strain was tested on *S. frugiperda* larvae *via* a highly concentrated suspension of $1 \times 10^8$ spores/mL. As shown in Fig. 3, the spray resulted in mortality rates of approximately $8.3 \pm 1.52\%$, $53.3 \pm 4.16\%$, and $69.66 \pm 2.50\%$ at 48, 96, and 144 h after *B. bassiana* infection (BbI), respectively.

## Overview of sequencing data

To gain a comprehensive understanding of *B. bassiana* CDL1, we conducted genome-wide transcriptome analysis *via* RNA-seq during its growth on 1/4-strength SDAY broth media and infection of *S. frugiperda* larvae at various time points. During fungal growth on 1/4-strength SDAY broth media, the total number of reads ranged from 36,164,664 to 49,431,470. The mapping rates and unique mapping percentages were consistently high, ranging between 90.30% and 91.52%. In contrast, when *B. bassiana* CDL1 infected *S. frugiperda* larvae, each treatment sample at different time points yielded varying numbers of raw reads. At 48 h post infection (hpi), the number of raw reads ranged from 39,080,418 to 40,853,568. At 96 hpi, the range was between 37,188,274 and 41,593,374 raw reads. At 144 hpi, the number of raw reads ranged from 44,245,422 to 47,535,454. At 48 and 96 hpi, only 1.30 to 1.90% of the reads uniquely mapped to the reference genome, indicating that *B. bassiana* CDL1 was present in lower quantities during the early stages of infection.

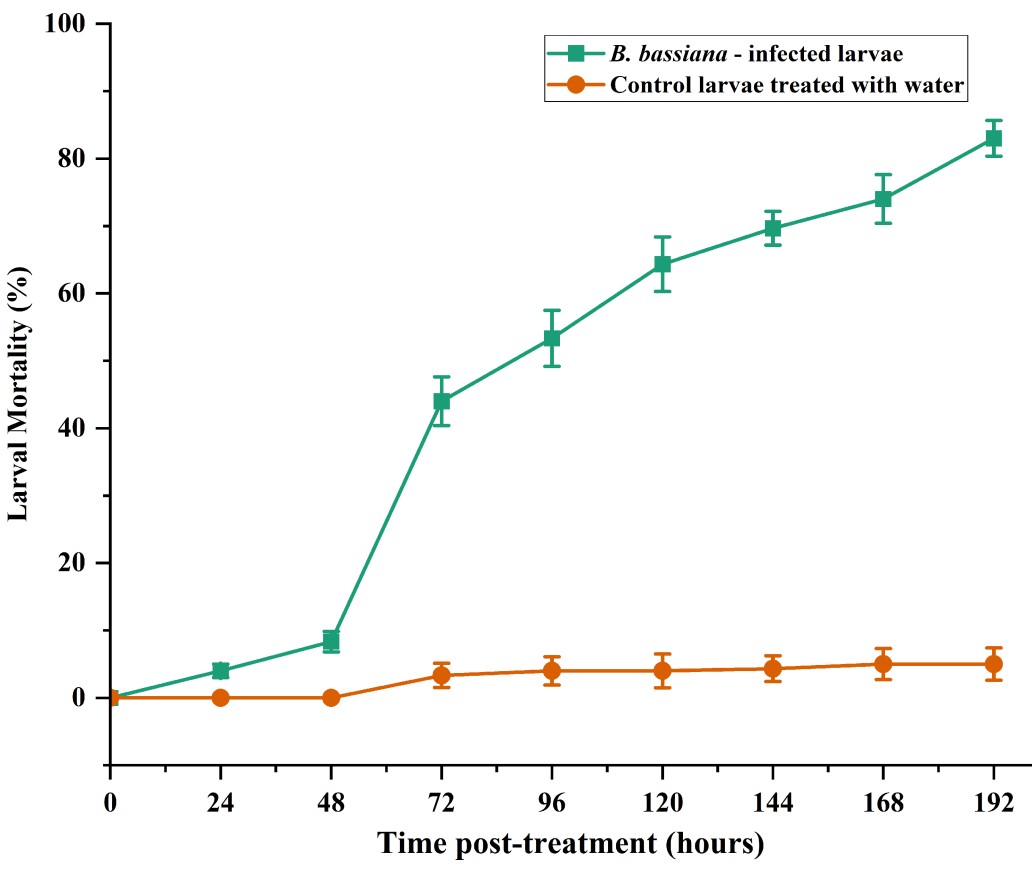

**Figure 3** **The mortality rates of *S. frugiperda* larvae infected with spore suspensions ($1 \times 10^8$ spores/mL) and water.** The error bars represent the standard error of the mean from three replicates.

However, at 144 hpi, 2.0 to 4.42% of the reads uniquely mapped to the reference genome, suggesting significant proliferation of the fungus at this stage. This increase in fungal-mapped reads indicates a higher fungal transcript abundance relative to earlier time points, reflecting active fungal growth and gene expression within the host. The elevated proportion of fungal reads suggests that the pathogen has successfully breached host defenses and is proliferating extensively, likely due to established colonization and tissue invasion at this stage of infection. These findings highlight the complexity and challenges of mapping reads in host–pathogen interactions. These findings also suggest potential variability in fungal gene expression, exhibiting significant changes during infection compared with that during growth in 1/4-strength SDAY broth media (Table 4).

**A comprehensive global analysis of the genes expressed during the growth of *B. bassiana* on artificial media and infection of *S. frugiperda***

Sequencing data from each sample were mapped to the *B. bassiana* genome, revealing that up to 5,225, 10,861, and 6,458 fungal genes were expressed in differential expression gene (DEG) libraries from *B. bassiana* growing on 1/4-strength SDAY broth media and infected larvae at 48, 96, and 144 h, respectively. These findings indicate that a significant

**Table 4** Summary of RNA-Seq data and mapping.

| Sample | Total read number | Unmapped (%) | Unique mapped (%) | Multiple mapped (%) | Total mapped (%) |
|---|---|---|---|---|---|
| F-48 h-a | 3,6164,664 | 9.36 | 90.38 | 0.26 | 90.64 |
| F-48 h-b | 3,8962,380 | 9.92 | 90.30 | 0.28 | 90.58 |
| F-48 h-c | 3,7013,128 | 9.32 | 90.41 | 0.27 | 90.68 |
| F-96 h-a | 4,1215,256 | 9.19 | 90.55 | 0.26 | 90.81 |
| F-96 h-b | 3,9500,932 | 9.26 | 90.49 | 0.24 | 90.74 |
| F-96 h-c | 3,9047,758 | 9.31 | 90.45 | 0.25 | 90.69 |
| F-144 h-a | 4,9431,470 | 8.78 | 90.87 | 0.35 | 91.22 |
| F-144 h-b | 3,8615,438 | 8.22 | 91.47 | 0.31 | 91.78 |
| F-144 h-c | 3,9765,996 | 8.20 | 91.52 | 0.28 | 91.80 |
| L-48 h-a | 4,0853,568 | 98.27 | 1.70 | 0.03 | 1.73 |
| L-48 h-b | 3,9080,418 | 98.31 | 1.67 | 0.02 | 1.69 |
| L-48 h-c | 3,9574,240 | 98.69 | 1.30 | 0.01 | 1.31 |
| L-96 h -a | 3,7188,274 | 98.18 | 1.80 | 0.02 | 1.82 |
| L-96 h-b | 4,1253,392 | 98.09 | 1.90 | 0.01 | 1.91 |
| L-96 h-c | 4,1593,374 | 98.11 | 1.87 | 0.02 | 1.89 |
| L-144 h-a | 4,4245,424 | 95.57 | 4.40 | 0.03 | 4.43 |
| L-144 h-b | 4,7535,454 | 97.95 | 2.01 | 0.04 | 2.05 |
| L-144 h-c | 4,4245,422 | 95.55 | 4.42 | 0.03 | 4.45 |

**Notes.**

F represents fungal growth in 1/4-strength SDAY broth medium, whereas L refers to larvae that are infected by the fungus; a, b, and c are three biological replicates.

number of genes in *B. bassiana* are activated during critical infection stages. For this study, differentially expressed genes (DEGs) were defined as those with a false discovery rate (FDR) < 0.05 and an absolute fold change $\geq$ 2. On the basis of these criteria, we identified 4,589, 5,839, and 6,458 DEGs between fungal growth on 1/4-strength SDAY broth media and infected larvae at 48, 96, and 144 h, respectively. Compared with those in larvae growing on 1/4-strength SDAY broth media, the numbers of upregulated genes in infected larvae were 930, 1,284, and 1,145 at 48, 96, and 144 h, respectively, whereas the numbers of downregulated genes were 3,659, 4,555, and 5,313 at these time points, respectively (Fig. 4). These findings suggest that each stage of infection influences distinct biological processes in the fungus.

## Enrichment of GO terms and KEGG pathway analysis

The GO annotation analysis of DEGs during *S. frugiperda* by *B. bassiana*, compared with that of the fungus growing on a 1/4-strength SDAY broth media, revealed significant enrichment in the following GO terms across all three GO domains: cellular process, metabolic process, binding, catalytic activity, and membrane, highlighting their critical roles in infection dynamics (Fig. 5). Among the top 20 significantly enriched GO terms, "catalytic activity" and "intrinsic component of membrane" were notably represented (Fig. 6). The increase in catalytic activity indicates the high involvement of DEGs in enzyme-mediated biochemical reactions, which are essential for the fungus to adapt to

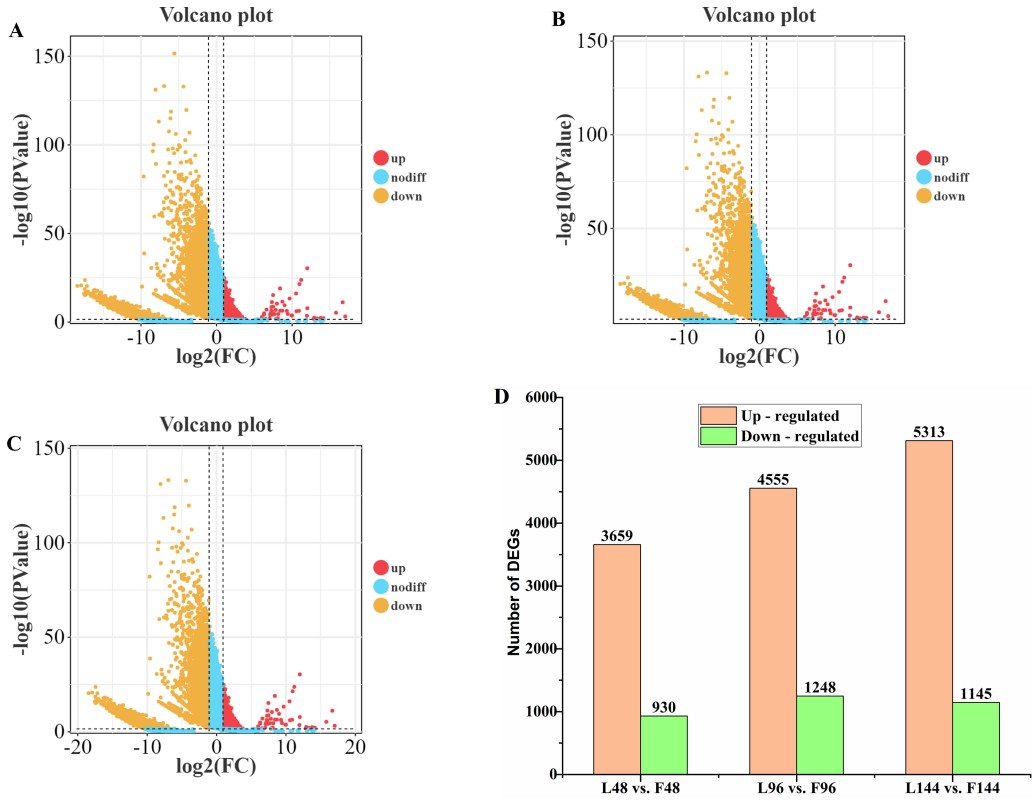

**Figure 4** **Volcano plots and differential gene expression (DEG) analysis of *B. bassiana* CDL1 during fall armyworm infection compared to growth on artificial medium.** (A) 48 h, (B) 96 h, (C) 144 h: Volcano plots showing the DEGs identified during infection compared to fungal growth on artificial medium. (D) Total number of DEGs at each time point ('L' = larvae infected by fungus, 'F' = fungus growth on artificial medium).

the host environment, acquire nutrients, and counteract host defenses. The enrichment of the intrinsic component of the membrane suggested that many DEGs are involved in membrane-related functions, such as transport, signaling, and maintaining cell integrity, which are crucial for pathogen entry into host cells, evasion of host immune responses, and intercellular communication during infection. To gain deeper insights into the functional roles of differentially expressed genes (DEGs) in *B. bassiana* during the infection of *S. frugiperda* and its growth on 1/4-strength SDAY broth media, KEGG pathway enrichment analysis was performed. This analysis categorized the biological functions of the DEGs by mapping them to terms in the KEGG database. Specifically, significant enrichment was detected in 25 pathways at 48 h post-infection, 18 pathways at 96 h, and 19 pathways at 144 hours' post-infection. The top 20 pathways, ranked by the gene ratio, are displayed as the ratio of the number of DEGs to the total number of genes in a given pathway (Fig. 7). Notably, at 48 hours' post-infection, metabolic pathways were among the top pathways. At 96 h, both metabolic pathways and ABC transporters were prominent. After 144 h, metabolic pathways and the biosynthesis of secondary metabolites were the

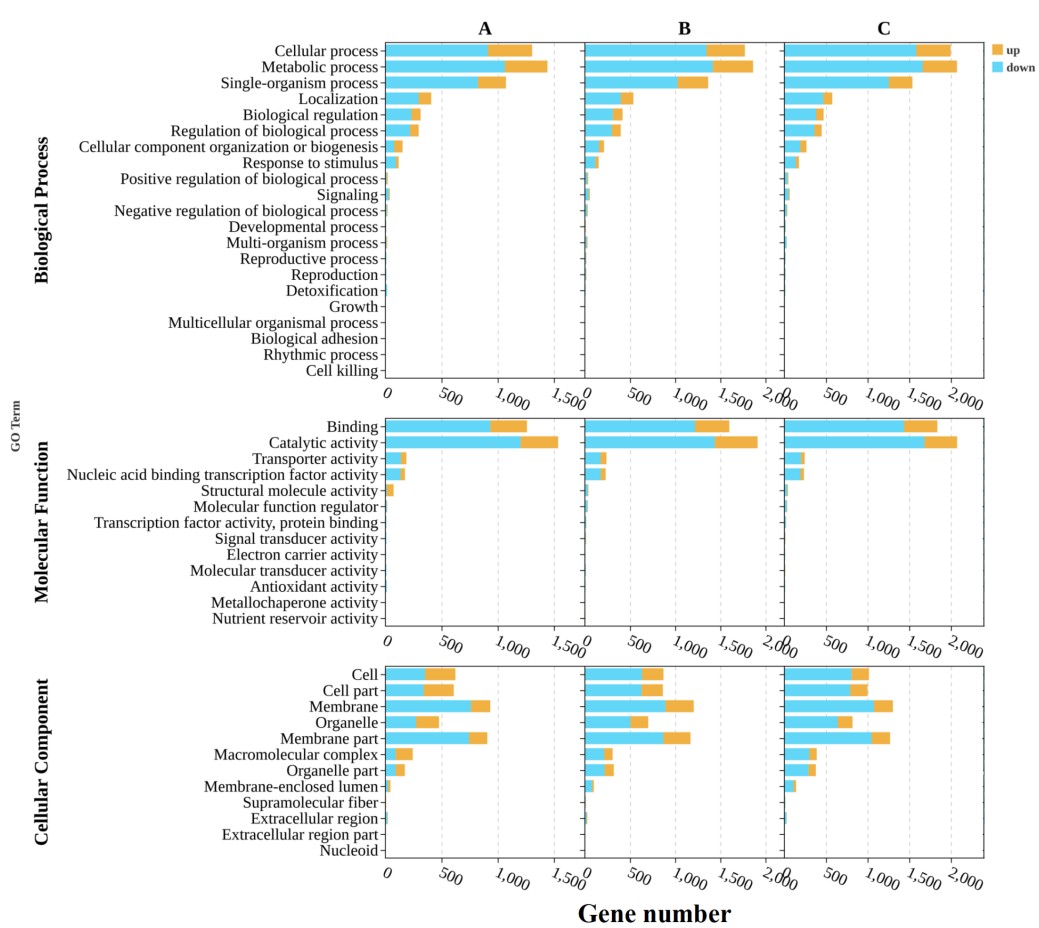

**Figure 5** Differential Gene Ontology (GO) analysis to compare the *B. bassiana* in fall armyworm larvae and growth on artificial media at different time periods. (A) The 48-hour timeframe; (B) the 96-hour timeframe; (C) the 144-hour timeframe.

top pathways. These results provide valuable insights into the temporal dynamics of gene expression and the key biological pathways involved in the infection process.

## Comparative temporal gene expression profiling of *B. bassiana* during infection of *S. frugiperda* larvae and growth on artificial media across different time points

Several genes exhibit significant upregulation in the infected condition (BbI) compared to growth on artificial media (BbM) at the 48-hour time point. Notably, the lipase gene shows a 10.46-fold increase in BbI, while the siderophore iron transporter gene mirB is upregulated by 11.48-fold in BbI. These findings suggest potential roles in lipid degradation and iron acquisition, both of which may contribute to fungal pathogenesis. By 96 h, the expression of the lipase gene decreases substantially, and mirB expression is also reduced, indicating a shift in metabolic processes as the infection progresses. In contrast, genes such as metalloprotease-like protein, cytochrome P450 3A9, and glutathione S-transferase omega-1 are significantly upregulated at later time points (96 and 144 h), suggesting

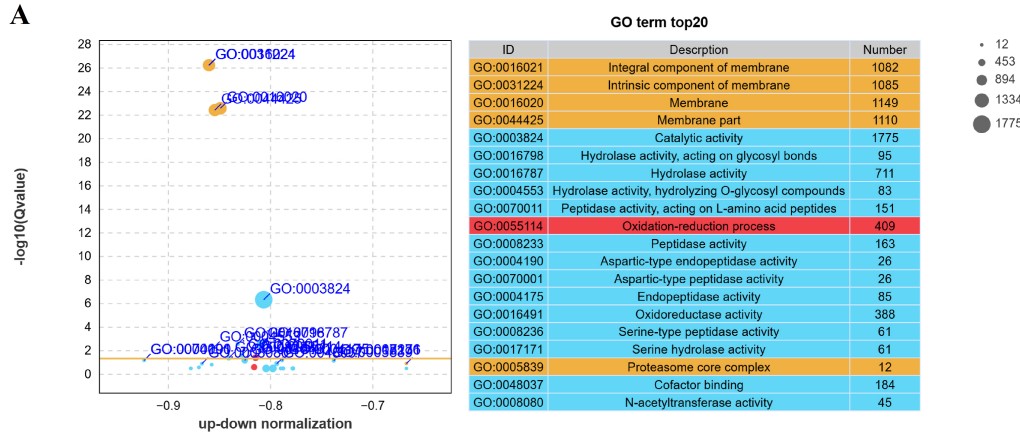

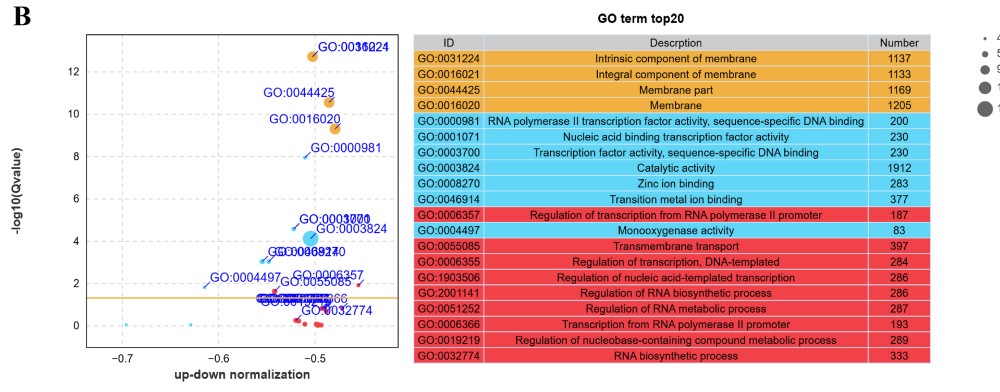

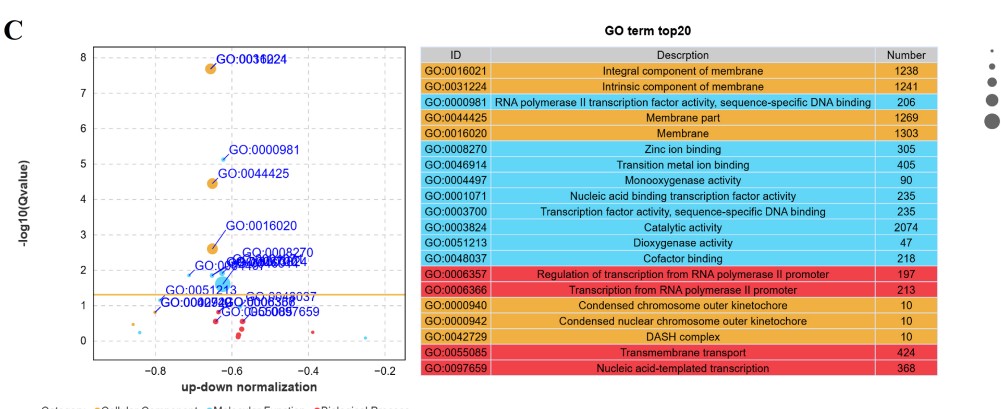

**Figure 6  Top 20 GO terms with significant enrichment.** (A–C) The top 20 Gene
Ontology (GO) terms with significant enrichment when the responses to fungal infection in larvae *versus*
fungal growth on artificial media at different timeframes were compared. (A) The 48-hour timeframe, (B)
the 96-hour timeframe, and (C) the 144-hour timeframe.

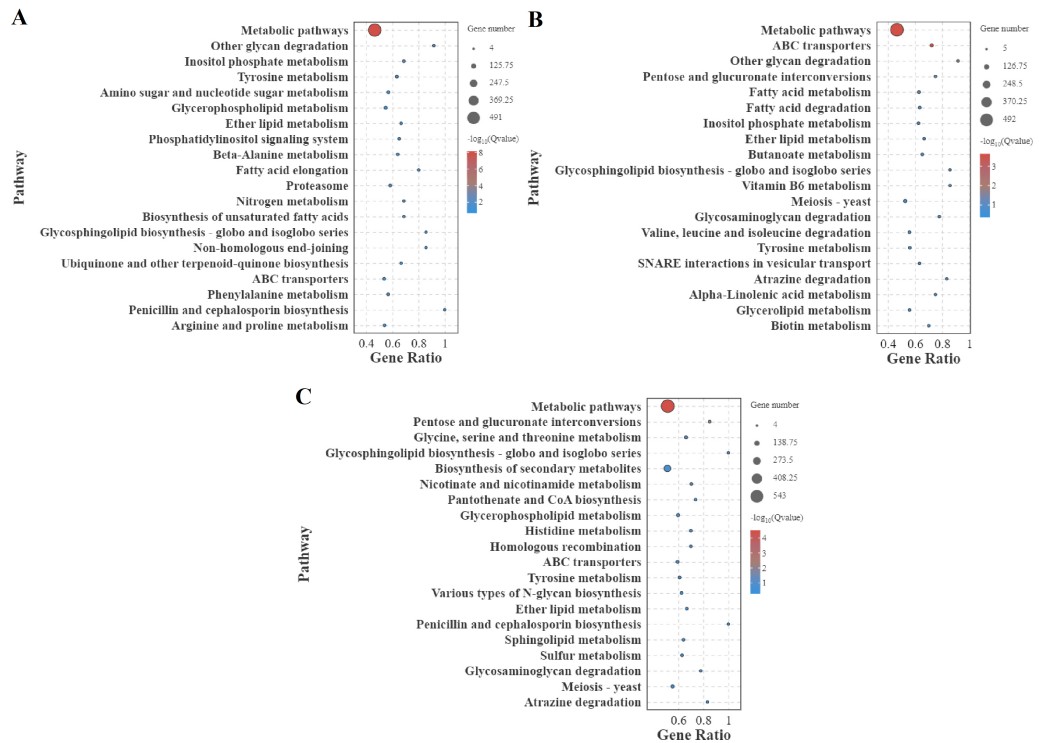

**Figure 7** **KEGG pathways for differentially expressed genes (DEGs) comparing infection in larvae and fungal growth on artificial medium.** (A) The 48-hour timeframe; (B) the 96-hour timeframe; (C) the 144-hour timeframe. The gene ratio (rich factor) is the ratio of the number of DEGs to the total number of genes in a given pathway, indicating the level of enrichment.

an increased reliance on proteolysis, detoxification, and secondary metabolism as the infection advances. Additionally, genes involved in stress response, such as ribonuclease and V-type ATPase, exhibit fluctuating expression profiles across the time points, reflecting the fungus's adaptation to host immune responses and environmental stresses. At 96 and 144 h, chitinase III and endochitinase III, which are associated with cell wall degradation, show upregulation, potentially facilitating the penetration of host tissues.

This dataset provides a comprehensive view of the temporal dynamics in gene expression of *B. bassiana* during host infection and growth on artificial medium, highlighting key metabolic and adaptive processes underlying fungal pathogenesis (Table 5 and Fig. 8).

## Validation of differentially expressed genes *via* RT-qPCR

To validate the differentially expressed genes (DEGs) identified through RNA-Seq analysis, we conducted RT-qPCR on 18 selected genes, comprising 10 randomly chosen DEGs and eight genes specifically associated with *B. bassiana* pathogenicity against *S. frugiperda*. The eight targeted genes—lipase (LIP), metalloprotease-like protein (MGG-80), siderophore iron transporter mirB (MFS2), sugar transporter STL1 (STL1), putative RING finger protein (SPCF3.16), cytochrome P450 3A9 (FUM15), serine/threonine-protein kinase rio2 (rio2), and chitinase III (chi2)—were selected based on the following criteria: (1)

**Table 5 Temporal gene expression profiling of *B. bassiana* during fall armyworm larval infection compared with that during growth on artificial media at different time points.**

| Gene name or description | Mean expression in BbI (48 h) (FPKM) | Mean expression in BbM (48 h) (FPKM) | log$_2$ Fold Change (48 h) | Mean expression in BbI (96 h) (FPKM) | Mean expression in BbM (96 h) (FPKM) | log$_2$ Fold Change (96 h) | Mean expression in BbI (144 h) (FPKM) | Mean expression in BbM (144 h) (FPKM) | log$_2$ Fold Change (144 h) |
|---|---|---|---|---|---|---|---|---|---|
| Lipase | 332.66 | 0.23 | 10.46 | 0.001 | 0.33 | −8.06 | 0.001 | 0.71 | −8.48 |
| ATP synthase subunit J | 1,762.16 | 442.22 | 2.00 | 606.62 | 288.44 | 1.07 | 166.94 | 343.37 | −1.35 |
| Siderophore iron transporter mirB | 573.24 | 0.20 | 11.48 | 0.001 | 0.78 | −9.61 | 0.001 | 4.30 | −12.07 |
| Sugar transporter STL1 | 923.17 | 14.86 | 5.95 | 48.30 | 38.20 | 0.34 | 22.58 | 23.70 | −0.06 |
| Putative RING finger protein | 15.07 | 10.64 | 5.66 | 70.20 | 82.58 | −0.23 | 22.58 | 23.70 | −0.06 |
| Extracellular serine-rich protein | 1.45 | 12.13 | −3.15 | 16.53 | 8.21 | 1.00 | 41.16 | 2.03 | 4.35 |
| Serine/threonine-protein kinase rio2 | 684.93 | 39.71 | 4.10 | 20.40 | 25.27 | −0.31 | 0.001 | 22.54 | −14.46 |
| Chitinase III | 1.27 | 7.75 | −2.60 | 27.32 | 3.81 | 2.84 | 20.30 | 11.20 | 0.86 |
| Endochitinase III | 0.001 | 3.97 | −11.95 | 24.33 | 4.17 | 2.54 | 51.22 | 41.88 | 0.29 |
| metalloprotease-like protein | 3.32 | 31.56 | −3.24 | 73.70 | 19.23 | 1.93 | 115.36 | 28.46 | 2.01 |
| Pyroglutamyl-peptidase 1 | 10.40 | 11.55 | −0.26 | 0.001 | 15.20 | −13.84 | 103.21 | 23.69 | 2.12 |
| Cytochrome P450 3A9 | 4.70 | 3.28 | 0.51 | 0.001 | 2.84 | −11.47 | 22.91 | 5.04 | 2.16 |
| putative methyltransferase C20orf7 | 44.87 | 16.06 | 1.48 | 0.001 | 11.70 | −13.51 | 106.64 | 23.04 | 2.21 |
| Glutathione S-transferase omega-1 | 0.57 | 14.87 | −4.70 | 0.001 | 11.49 | −13.48 | 79.17 | 7.84 | 3.33 |
| beta-lactamase/ transpeptidase-like protein | 0.89 | 3.07 | −1.77 | 0.001 | 3.25 | −11.66 | 19.21 | 2.75 | 2.80 |
| ubiquinol-cytochrome C reductase | 10.60 | 37.56 | −1.82 | 0.001 | 23.91 | −14.54 | 299.54 | 26.03 | 3.52 |
| Ribonuclease | 0.001 | 29.63 | −14.85 | 0.001 | 39.15 | −15.25 | 97.06 | 16.44 | 2.56 |
| Fructosamine-3-kinase | 7.19 | 187.47 | −4.70 | N/A | N/A | N/A | 200.03 | 22.92 | 3.12 |
| aldo/keto reductase | 725.13 | 197.91 | 1.87 | N/A | N/A | N/A | 105.44 | 7.59 | 3.79 |
| V-type ATPase | 0.001 | 0.14 | −7.16 | 15.60 | 1.07 | 3.86 | 0.001 | 0.758 | −9.56 |
| Catalase-1 | 3.36 | 3.03 | 0.14 | N/A | N/A | N/A | 92.50 | 4.80 | 4.26 |
| Choline-sulfatase | 19.41 | 13.63 | 0.50 | 67.01 | 10.14 | 2.72 | 0.001 | 16.53 | −14.01 |

**Table 5** (*continued*)

| Gene name or description | Mean expression in BbI (48 h) (FPKM) | Mean expression in BbM (48 h) (FPKM) | log₂ Fold Change (48 h) | Mean expression in BbI (96 h) (FPKM) | Mean expression in BbM (96 h) (FPKM) | log₂ Fold Change (96 h) | Mean expression in BbI (144 h) (FPKM) | Mean expression in BbM (144 h) (FPKM) | log₂ Fold Change (144 h) |
|---|---|---|---|---|---|---|---|---|---|
| Putative sucrose utilization protein SUC1 | 779.71 | 54.56 | 3.83 | N/A | N/A | N/A | 291.91 | 189.71 | 0.62 |
| Alkylated DNA repair protein alkB 8 | 26.41 | 33.62 | −0.34 | 0.001 | 15.27 | −13.84 | 66.16 | 11.14 | 2.56 |
| Putative acyl-CoA synthetase YngI | 73.54 | 0.001 | 16.16 | N/A | N/A | N/A | 0.001 | 0.07 | −6.29 |
| Putative metal ion transporter C17A12.14 | 589.31 | 49.60 | 3.56 | N/A | N/A | N/A | 57.29 | 71.56 | −0.32 |
| ppic-type ppiase domain containing protein | 48.73 | 22.97 | 1.08 | 0.001 | 21.31 | −14.37 | 216.66 | 24.64 | 3.13 |
| Sexual differentiation process protein isp4 | 7.84 | 7.91 | −0.01 | 0.001 | 7.58 | −12.88 | 56.21 | 11.17 | 2.33 |

**Notes.**

The log₂ ratio (BbI/BbM) represents the fold change in gene expression between two conditions: BbI (*B. bassiana*-infected larvae) and BbM (*B. bassiana* growth on 1/4-strength SDAY broth medium). N/A indicates data for genes were not detected for the given time point or condition. The FPKM method is employed to eliminate the influence of different gene lengths and sequencing discrepancies on the calculation of gene expression.

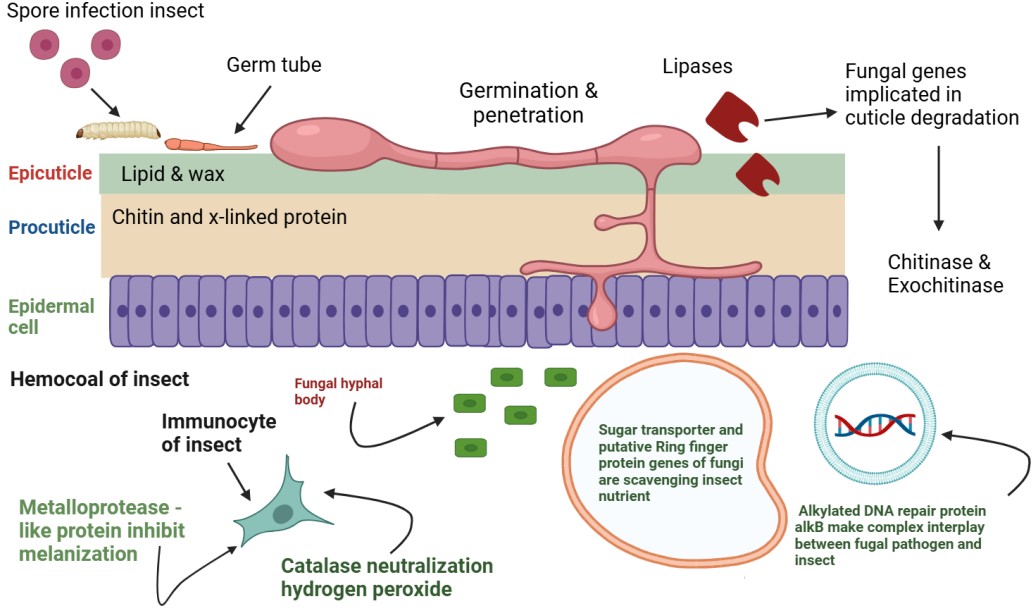

**Figure 8** Temporal dynamics of virulence related gene expression in infection of *B. bassiana* to *S. frugiperda*.

They were among the most significantly differentially expressed; (2) they represent key pathways involved in pathogenicity, such as cuticle degradation, nutrient acquisition, stress responses, and detoxification; and (3) they include both up- and down-regulated genes across different time points of infection (48 h, 96 h, and 144 h). The RT-qPCR results confirmed the differential expression patterns observed in the RNA-Seq data for all 18 genes (Fig. 9). Among the eight targeted genes, significant upregulation was observed at early stages of infection (48 h) for genes such as lipase, siderophore iron transporter mirB, and sugar transporter STL1, which are involved in cuticle degradation and nutrient acquisition. In contrast, genes like metalloprotease-like protein and chitinase III exhibited dynamic expression patterns, with upregulation at later stages (96 h and 144 h), suggesting their roles in host tissue degradation and nutrient acquisition during advanced infection. Genes associated with stress responses (putative RING finger protein, serine/threonine-protein kinase rio2) and detoxification (cytochrome P450 3A9) showed variable expression across time points, reflecting their potential roles in adapting to host defenses and environmental stress. The correlation between RNA-seq and RT-qPCR results was strong across all time points, confirming the reliability of the RNA-seq data (Fig. 9). These findings validate the differential expression of genes involved in key pathogenic pathways and provide further insights into the temporal regulation of *B. bassiana*'s infection process in *S. frugiperda*.

## DISCUSSION

In this study, we investigated the effects of *B. bassiana* CDL1 on *S. frugiperda* and compared the gene expression profiles of the fungus during infection with those observed under standard growth conditions. Our analysis provides insight into how *B. bassiana* alters its gene expression in response to its insect host and highlights the differential expression of genes involved in pathogenicity and the stress response.

The mortality of the larvae observed in our study significantly influenced overall mortality rates and exhibited a similar trend. We found that the percentage of larval mortality was positively correlated with total mortality, and this correlation became stronger as the concentration of *B. bassiana* spores increased. These findings were supported by *Nelly, Reflinaldon & Meriqorina (2023)*, who reported a positive correlation between *B. bassiana* concentration and *S. frugiperda* larval mortality. Specifically, they reported that a *B. bassiana* suspension with a concentration of $1 \times 10^9$ conidia/mL resulted in the highest mortality rate of 84%. These findings suggest that the effectiveness of *B. bassiana* in infecting *S. frugiperda* larvae increases with increasing conidia density. The death of larvae caused by entomopathogenic fungi is a result of the production of toxic metabolites, including beauvericin, beauverolite, bassianalite, bassianolide, and isorolite (*Abdullah, Abd El-Wahab & Abd El-Salam, 2024*; *Bi et al., 2023*; *San Juan-Maldonado et al., 2024*; *Vishaka et al., 2020*). These compounds destroy the digestive system, muscles, nervous system, and respiratory system of insects; the synergy of all these toxins causes the death of the preyed-upon insect (*Pedrini, 2022*). These findings are consistent with earlier research on other entomopathogenic fungi, further underscoring the potential of *B. bassiana* as an effective biocontrol agent.

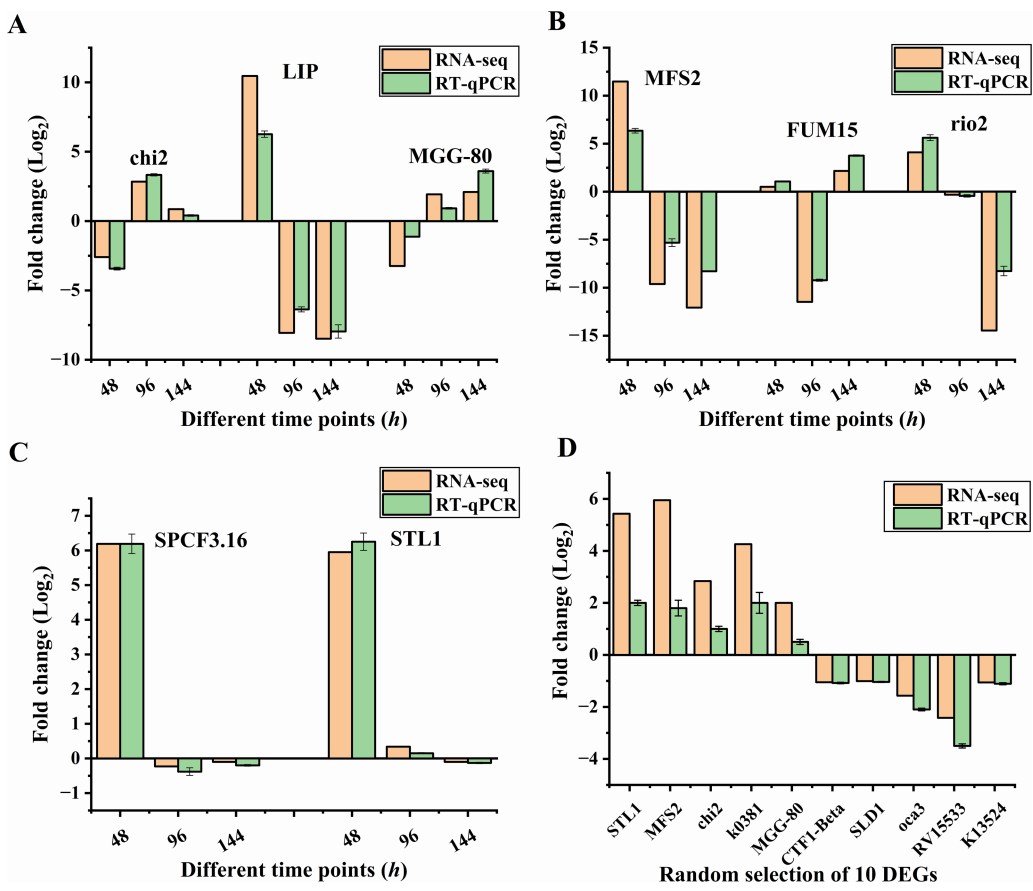

**Figure 9** **Validation of differentially expressed genes (DEGs) by RT-qPCR.** (A–C) Expression profiles of eight targeted genes associated with *B. bassiana* pathogenicity against *S. frugiperda* across different time points (48 h, 96 h and 144 h). (D) Expression profiles of 10 randomly selected DEGs. Error bars in all panels (A–D) represent mean ± SE from three biological replicates.

On the other hand, our results demonstrate that the latent effect of *B. bassiana* CDL1 spores during the larval stage can extend to the pupal and adult stages. This is evidenced by the observed pupal mortality in *S. frugiperda* insects treated with spores. These findings are consistent with those of *Islam et al. (2023)*, who reported that the EPF isolate TGS2.3 had a sublethal effect on different life stages of *Spodoptera litura*. Specifically, fungal infection during the larval stage directly results in pupal mortality. The entomopathogenic fungus *B. bassiana* reportedly interferes with insect molting, preventing the larvae from successfully progressing into the pupal stage (*Torrado-León, Montoya-Lerma & Valencia-Pizo, 2006*). The molting process is fundamental to the production of new cuticles, but it can be disrupted by fungal impact. Since the formation of these cuticles relies on nutritional resources, any imbalance in hemolymph nutrients caused by fungal infection can negatively impact the molting process at different stages (*Islam et al., 2023*). The findings of the present study suggest that the sex ratio of adult *S. frugiperda* is significantly biased toward females due to the treated larvae. This indicates that male individuals within

the population are more susceptible to *B. bassiana* spores than females. In their study, *Korany et al. (2019)* reported that, compared with female chitinase, crude chitinase had a greater effect on the growth of male individuals during their developmental stages. Several factors can contribute to sex differences in disease susceptibility, such as variations in body mass and the immune response. Generally, males are more susceptible to diseases, whereas females often have stronger immune responses. This disparity in immune strength may explain the increased incidence of autoimmune diseases and malignancies among females (*Kecko et al., 2017*). The dominance of emerging adult females after treatment was clear. However, their fecundity study revealed no eggs at a relatively high concentration of 1 $\times$ $10^6$ spores/mL. The concentration-dependent reduction in fecundity aligns with the findings of *Kaur et al. (2014)*. These authors also reported significant reductions in *S. litura* fecundity, adult emergence, longevity, and hatching percentages when *S. litura* was exposed to relatively high concentrations of secondary metabolites from *Streptomyces hydrogenans* DH16. Furthermore, they observed morphological abnormalities compared with those in the control groups. The ability of *B. bassiana* to reduce *S. frugiperda* female fecundity and survival is attributed to physiological alterations resulting from pathogenic infection (*Jin et al., 2015*; *Usman et al., 2021*). For example, entomopathogenic fungi (EPFs), such as *B. bassiana,* can deplete sugar and other compounds in insect hemolymph (*Jin et al., 2015*; *Peng et al., 2015*; *Xia, Clarkson & Charnley, 2002*), significantly affecting host insect fitness parameters (*Jin et al., 2015*). The decrease in oviposition rate and fertility among infected females is closely linked to EPF action, which negatively impacts insect populations and compensates for their delayed mortality (*Dimbi, Maniania & Ekesi, 2013*). Furthermore, the antifeedant activity of *B. bassiana* during the invasive process may contribute to the observed reduction in fecundity (*Ekesi, 2001*). These sublethal effects are critical for long-term pest suppression, as they disrupt the reproductive potential of surviving populations.

Our transcriptomic analysis of *B. bassiana* CDL1 during its interaction with *S. frugiperda* reveals significant changes in gene expression compared to its growth in artificial media. Over time, the increase in uniquely aligned reads indicates a rise in fungal proliferation within the host. These findings align with the results of *Chen et al. (2018a)*, who observed a similar increase in aligned reads over time during the infection of *B. bassiana* in *Galleria mellonella*. Similarly, *Zhou et al. (2019)* reported an increase in read mapping during the infection of *B. bassiana* against both *G. mellonella* and *Plutella xylostella*, further supporting the observed trend of enhanced fungal activity during host infection.

Temporal gene expression profiling identified specific genes with significant expression changes during infection compared to growth on artificial media across different time points. In the early stages of infection, adhesion to the host cuticle and subsequent penetration are critical for the successful invasion of *B. bassiana* CDL1. Understanding the differential expression of these genes provides valuable insights into the infection process. For instance, serine/threonine-protein kinases (STPKs) play a pivotal role in host recognition and the activation of downstream signaling pathways, highlighting their importance in fungal pathogenesis. Specifically, the rapid upregulation of STPK gene expression at 48 hours' post-infection (BbI), followed by downregulation at 96 h and

sustained low levels thereafter, suggests their involvement in the formation of infection structures. This observation is consistent with the findings of *Gormal et al. (2024)*, who demonstrated that STPKs are activated upon detecting host-specific receptors or molecules, triggering signal transduction pathways such as mitogen-activated protein kinase (MAPK) and protein kinase A (PKA). During cuticle penetration, *B. bassiana* upregulates genes encoding lipases and chitinases, which target the epicuticular lipids and chitin layers (*Gao et al., 2011*; *Zhang et al., 2012*). Notably, at 48 hours' post-infection (BbI), we observed a marked increase in lipase expression, suggesting its key role in degrading host lipids to facilitate penetration. Comparing these findings with studies on other fungal entomopathogens, such as *Metarhizium* and *Isaria*, reveals both similarities and differences in their gene expression profiles. For instance, *Metarhizium* species show an upregulation of lipases during the early stages of infection, emphasizing the role of lipid degradation in penetrating the host cuticle and evading immune defenses (*Gao et al., 2011*; *Reingold et al., 2024*). In contrast, *Isaria* species may regulate proteolytic enzymes differently during these early stages, reflecting distinct strategies for overcoming host immune responses. Similarly, the upregulation of chitinases is noteworthy, as these enzymes are essential for degrading the chitin matrix, further facilitating successful cuticle penetration. Our results align with those of *Lai et al. (2017)*, who found that chitinases were highly expressed at 60 h post-infection (hpi), promoting the digestion of mosquito cuticle chitin and enabling penetration of the procuticle. This underscores the pivotal role of these enzymes in the initial interaction between fungal pathogens and their insect hosts. Our results also show high expression of the catalase gene after 144 h of infection, likely reflecting a response to host-derived oxidative stress, as reactive oxygen species (ROS) such as hydrogen peroxide ($H_2O_2$) are produced to combat pathogens. Catalase, an antioxidant enzyme, breaks down $H_2O_2$, protecting the pathogen from oxidative damage. This delayed upregulation suggests a strategic adaptation to establish infection and counteract sustained oxidative stress during later stages. Similar findings were reported by *Wang et al. (2013b)*, who demonstrated increased catalase expression in prolonged host-pathogen interactions, highlighting its role in pathogen survival under oxidative stress conditions. Furthermore, our data reveal that the RING finger protein exhibits increased expression at 48 hpi, suggesting its involvement in protein degradation, signaling, and fungal adaptation. This aligns with previous studies by *Zhang et al. (2010)*, who reported similar roles for RING finger proteins in fungal pathogens, highlighting their importance in modulating host interactions and stress responses. Similarly, the upregulation of the sugar transporter STL1 in our study underscores *B. bassiana*'s enhanced ability to exploit host sugars, which is consistent with the findings of *Wang et al. (2013a)*. These authors demonstrated that STL1 facilitates fungal proliferation by optimizing nutrient acquisition from the host, further supporting our observations. Additionally, the increased expression of the siderophore iron transporter (mirB) at 48 hpi in our study suggests a critical role in iron acquisition, which is essential for fungal virulence. This finding corroborates earlier work by *Wang et al. (2017)*, who identified mirB as a key player in iron scavenging, a process vital for pathogen survival and host colonization. The upregulation of metalloprotease-like genes at 144 hours' post-infection (hpi) suggests their potential role in suppressing the host immune

system, possibly by inactivating prophenoloxidase, a key component of insect immunity. This observation aligns with the findings of *Huang et al. (2020)*, who highlighted the crucial function of metalloproteases in evading host defenses. Likewise, the increased expression of cytochrome P450 3A9 (CYP3A9) at 144 hpi indicates its involvement in detoxification processes. This is consistent with studies by *Črešnar & Petrič (2011)* and *Forlani et al. (2014)*, which identified cytochrome P450 as essential for fungal survival, particularly in detoxifying xenobiotics, breaking down harmful compounds, and synthesizing secondary metabolites vital for pathogenicity.

## CONCLUSIONS

This study highlights the virulence of *B. bassiana* CDL1 against *S. frugiperda*, demonstrating a concentration-dependent effect on mortality, development, and reproduction. Higher spore concentrations significantly increased mortality rates, delayed developmental progression, and reduced fecundity, culminating in complete reproductive suppression at the highest concentrations. The pathogenic potential of *B. bassiana* CDL1 was evident through its ability to disrupt multiple life stages of the insect.

Transcriptome analysis revealed distinct gene expression patterns during fungal infection and artificial growth, identifying key pathways associated with virulence, metabolism, and host adaptation. These findings underscore *B. bassiana* CDL1 as a highly promising biocontrol agent, exhibiting strong pathogenicity against *S. frugiperda*.

### Funding

This study was supported by the National Key R&D Program of China (2022YFD1401001) and the Science and Technology Innovation Project of the Qinling Institute in NWAFU (2452023301). The funders had no role in study design, data collection and analysis, decision to publish, or preparation of the manuscript.

### Grant Disclosures

The following grant information was disclosed by the authors:
National Key R&D Program of China: 2022YFD1401001.
Science and Technology Innovation Project of the Qinling Institute in NWAFU: 2452023301.

### Competing Interests

The authors declare there are no competing interests.

### Author Contributions

- Hamdy H. Aly performed the experiments, analyzed the data, prepared figures and/or tables, authored or reviewed drafts of the article, and approved the final draft.
- Yun Meng performed the experiments, authored or reviewed drafts of the article, and approved the final draft.

- Dun Wang conceived and designed the experiments, authored or reviewed drafts of the article, and approved the final draft.

## DNA Deposition

The following information was supplied regarding the deposition of DNA sequences:

The raw sequence reads are available at GenBank: PRJNA1106630.

## Data Availability

The raw data is available in the Supplemental Files.

## Supplemental Information

Supplemental information for this article can be found online at http://dx.doi.org/10.7717/peerj.19591#supplemental-information.

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
