# Peer review of "Comparative gene expression analysis of Beauveria bassiana against Spodoptera frugiperda"

_PeerJ, doi:10.7717/peerj.19591_

## Round 0.1 · original submission · Major Revisions

The reviewers acknowledge this paper's values but also raise a number of criticisms for improvement. Please revise the paper according to their comments.

Reviewer 1 ·

Basic reporting

The study provides valuable insights into the molecular mechanisms underlying B. bassiana infection dynamics.
The combination of bioassays and transcriptomic analysis enhances the depth of the study.

Language and grammar could be improved for better clarity and readability.
Enhance figure legends and data visualization for improved clarity.

Experimental design

Additional methodological details and more detailed results could be provided for better reproducibility.

Validity of the findings

Strengthen the discussion by comparing findings with previous studies and providing practical implications for biocontrol.

Additional comments

Enhance figure legends and data visualization for improved clarity.

·

Basic reporting

Thank you for your opportunity to review this paper. Paper looks good and have broad range significance in the field.
Here please find my suggestions and comments
Major
Comment 1. Why author write this paper?
Comments 2. What will be contribution of this biocontrol to society? If this paper contribute to develop a novel protocol regarding biocontrol of pathogen then why this study is limited to only larva? It should be expand fully to anti bacterial and antifungal activities.
Minor comment
Line 46-47: please remove the key words that already used in title l.
Line 53-53: author should double check the number of species of 2024. It already reached upto 1600 or more.
Line 81-82: please revise these lines as not clearing the concept.
Line 110-111: Please clear this point for purpose to add tween 20?
Figure 3: please make horizontal and vertical title consistent.

Experimental design

Experimental design need to improve

Validity of the findings

This paper is according to aims and scop of journal

Additional comments

This paper will have good impact on readers or researchers that will lead further research

Reviewer 3 ·

Basic reporting

This manuscript presents a study investigating the molecular mechanisms of Beauveria bassiana pathogenicity against the fall armyworm (Spodoptera frugiperda) through comparative transcriptomic analysis. The authors compared gene expression profiles of B. bassiana during infection of S. frugiperda larvae and during growth on artificial media at different time points. The study also includes bioassays to assess the insecticidal efficacy of different spore concentrations. The research addresses a relevant topic in biocontrol and provides valuable data on the gene expression dynamics of B. bassiana during host infection. However, the manuscript has some weaknesses, particularly in the validation of transcriptomic data and the depth of discussion. The most significant concern is the qPCR validation, where the random selection of genes weakens the support for the RNA-Seq findings. A more targeted and logically justified gene selection for qPCR validation is crucial. Furthermore, several sections require refinement to improve clarity, update references, and strengthen the overall impact of the study.

Experimental design

The most significant concern is the qPCR validation, where the random selection of genes weakens the support for the RNA-Seq findings. A more targeted and logically justified gene selection for qPCR validation is crucial.

Validity of the findings

See the additional comments for detail.

Additional comments

1. Line 50-74: Many cited references in the introduction, especially in the background section, are outdated. Please replace these with more recently published and impactful studies in the field of entomopathogenic fungi and transcriptomics to reflect the current state of knowledge. For example, in line 70, replace the cited references with the newly published one doi: 10.1111/1749-4877.12725
2. The font size in Figures 4 and 5 is too small and difficult to read. Please increase the font size for better readability.
3. Figure 1 should include scale bars to provide a sense of size for the morphological changes observed.
4. Line 86: The insect Spodoptera frugiperda deserves a brief introduction in the introduction section, highlighting its significance as an agricultural pest to further justify the study’s relevance.
5. Line 99: While a reference is cited for the artificial diet, a brief illustration of the diet composition is essential for readers to understand the rearing conditions without having to look up the cited paper.
6. Line 114: More details are needed for the spray method. Specifically, mention the volume of spore suspension sprayed per larva or per unit area and describe the method of ensuring uniform spore coverage.
7. Line 140: Clarify the rationale for using a high concentration of 1 x 108 spores/mL to determine differential gene expression. Explain why this specific concentration was chosen for the transcriptomic analysis and if it represents a biologically relevant infection scenario.
8. Line 195: The random selection of 10 DEGs for RT-qPCR validation is a significant weakness. Random selection is not a good option for validation. Add an additional qPCR validation of representative DEGs using a more robust and logically justified gene selection strategy. Consider selecting genes that are: among the most significantly differentially expressed, representing key pathways, known to be involved in pathogenicity, and include both up and down-regulated genes across different time points. Clearly state and support the gene selection principle in the methods.
9. Line 549: The conclusion that the findings inform more effective biocontrol strategies is overstated and not fully supported by the current results. Tone down or qualify this conclusion as the current results do not directly translate to field applications without further validation.
10. Lines 341-370: The “Comparative temporal gene expression profiling…” section mixes results and discussion. Focus on presenting results in this section and move interpretive discussion to the Discussion section.
11. Line 377: Maintain consistency in using either the Latin name (Spodoptera frugiperda) or the common name (fall armyworm) of the pest throughout the manuscript.
12. Line 380: Rephrase the sentence using “adapt” as it may not be the most appropriate word. Consider alternatives like “modulates” or “alters” gene expression in response to or during host infection.
13. Lines 382-383: Rephrase the sentence “The mortality of the larvae observed in our study had a significant effect on overall mortality rates and followed the same trend” for clarity. Explain the intended meaning.
14. Lines 390-391: Add references to support the statement about toxic metabolites causing insect death. Cite relevant reviews or studies on Beauveria bassiana toxins. For example, cite doi: 10.1111/1749-4877.12867 or similar articles.
15. Streamline the discussion section to be more focused and concise. Primarily address the key findings of the study and their direct implications. Reduce lengthy general descriptions and focus on interpreting the specific results of this transcriptomic analysis. Avoid over-speculation. The discussion section is currently too long. Please shorten and make it more focused.

---

## Round 0.2 · Minor Revisions

The reviewers have acknowledged that your paper has improved significantly, but they still think it needs some work. Fortunately, one of the reviewers has suggested some revisions. Please see the annotated manuscript for the reviewer's suggestions for revisions. I hope the reviewer's suggestions are helpful to further improve your manuscript.

Reviewer 1 ·

Basic reporting

It is now improved in language and some clarity in results, but it needs to add the comments given by reviewers.

Experimental design

It ok now

Validity of the findings

Results need to check as per materials and methods.

Additional comments

Do correction as per suggestions.

Annotated reviews are not available for download in order to protect the identity of reviewers who chose to remain anonymous.

---

## Round 0.3 · accepted · Accept

The reviewers and I confirm that the authors have addressed all of the reviewers' concerns. Therefore, I am happy to accept this paper for publication in PeerJ. Now, this paper is ready for publication.

Reviewer 1 ·

Basic reporting

It is now improved and very clear and useful for readers.

Experimental design

Now all experimental design, procedure and information are clear.

Validity of the findings

It certainly has impact and useful to develop a potential bio-insecticide based on these results.

Additional comments

It is now OK and can be publish.

·

Basic reporting

I accept this paper in this form

Experimental design

Fine

Validity of the findings

Fine

Additional comments

English editing is needed to improve further.